# The Performance of Environmental and Health Impact Assessment Implementation: A Case Study in Eastern Thailand

**DOI:** 10.3390/ijerph21050644

**Published:** 2024-05-18

**Authors:** Pattajaree Krasaesen, Vilas Nitivattananon, Malay Pramanik, Joyee Shairee Chatterjee

**Affiliations:** 1Urban Innovation and Sustainability, Department of Development and Sustainability, School of Environment Resources and Development, Asian Institute of Technology, Klong Luang 12120, Pathum Thani, Thailand; malay@ait.asia; 2Faculty of Public Health, Mahidol University Amnatcharoen Campus, Muang 37000, Amnat Charoen, Thailand; 3Gender and Development Studies, Department of Development and Sustainability, School of Environment Resources and Development, Asian Institute of Technology, Klong Luang 12120, Pathum Thani, Thailand; joyeec@ait.asia

**Keywords:** environmental health impact assessment (EHIA), EIA performance, environmental statement review, Eastern Thailand, strategic environmental assessment

## Abstract

Environmental impact assessment (EIA) performance has remained of interest, and over the past ten years, the evaluation technique has evolved. Thailand implemented an EIA with a health impact assessment (HIA) as an environmental health impact assessment (EHIA), which necessitated investigating and developing these instruments; however, its implementation performance has been questioned. The main goal of this study is to comparatively assess how well EIAs and EHIAs are performed in projects in an area in Thailand. Six projects in various sectors that were implemented in Eastern Thailand were studied. The 162 residents (nine local authorities and 153 villagers) closest to the project completed a survey and evaluated the performance according to three aspects (i.e., substantive, procedural, and transactive), using a rating scale and evaluation checklists. The results were presented as a percentage of the total scores and interpreted according to the five scales. The overall performance reached a satisfactory level, albeit not significantly different between cases; however, it was pointed out that the shortcomings of EHIAs and EIAs, particularly their dependability, lack of public involvement, and the need for more transparency, could be addressed through the establishment of an open access database, which would help to simplify the assessment of all stages of EIAs and EHIAs.

## 1. Introduction

An environmental impact assessment (EIA) is a mechanism used to guarantee that decision-makers examine the environmental implications of a project before determining whether or not to proceed. An EIA aims to integrate environmental considerations and support into a decision-making system, avoiding or reducing negative impacts and thus also protecting natural resources and ecological processes to promote sustainable development [1]. Nowadays, impact assessments have emerged as a primary practice in developing countries, such as the Philippines, Myanmar, Cambodia, Vietnam, and the Lao PDR [2,3,4]. Since the EIA system was established in 1969, academicians and researchers have continuously scrutinized its efficacy in terms of both theoretical frameworks and practical applications [5,6]. Thailand has implemented EIAs for more than 40 years for projects that could cause major environmental impacts. In Thailand, a health impact assessment forming a part of an environmental impact assessment, called an “EHIA”, has been mandatory since 2009 [7] and is needed for projects expected to have significant impacts on society, health, or the environment. An EHIA necessitates comprehensive health information and evidence connected to quality of life values. This information is crucial for assessing the interconnections and potential effects of any development on the physical, mental, and spiritual well-being of vulnerable groups.

In Thailand, EHIA processes are conducted at various stages and levels, ranging from project proposals to operational phases, and extending across local, national, or even international scopes. These assessments are essential for diverse undertakings, including mining projects, trade agreements within commodity supply chains, and expansions in industrial estates. Currently, 35 types of activities/projects require EIAs, whereas 11 require EHIAs; however, despite continued efforts to improve these structures, social disputes continue to grow. This suggests that the fundamental issues have not been fully addressed, and that systems are constrained by limits that reduce their effectiveness. As a result, they are unable to play their intended role as effective tools for preventing and mitigating environmental and health consequences.

The question of the performance and quality of EIAs/EHIAs has arisen, and research conducted to find advantages and disadvantages or weaknesses in each context for the countries is increasing. Similarly, in Thailand, the performance of EIAs/EHIAs in practice is still ambiguous.

Thailand has seen many cases [8,9,10,11] in which complaints about the inadequate performance of EIAs and EHIAs were noted. In addition, the lack of clear definitions of EIA laws in Thailand, especially lists of the types and scales of projects, have created problems in the interpretation and performance of EIAs and EHIAs [8].

However, previous quality evaluation tools have been assessed only in environmental assessment reports and have used academics, practitioners, consultant firms, and EIA and EHIA professionals as the evaluators, while the public or community, who are the direct stakeholders of EIA and EHIA processes, were neglected. Therefore, this research evaluates the performance of implementing EIAs and EHIAs, covering all stages of EIA and EHIA processes, including the impact assessment, decision-making, and monitoring, through developed evaluation checklists; changes the evaluators to be the key stakeholders in an area, such as residents and local authorities, including civil organizations, health organizations, and environmental organizations; and identifies the factors related to performance.

## 2. Literature Review

### 2.1. Environmental Impact Assessment

An EIA is a process that, prior to important choices and commitments being made, identifies, predicts, evaluates, and mitigates the biophysical, social, and other pertinent implications of development projects [12]. The main advantages and benefits of an EIA are the information on improved project designs, the opportunities for public involvement, increased accountability and transparency during the development process, and also more informed decision making.

### 2.2. Dimensions for Evaluating Environmental and Health Impact Assessment Performance

The performance of an impact assessment system can be defined as the extent to which it works, its findings, and whether it gains the acceptance of key stakeholders. Stakeholders can also gain information and experience that influence their decision-making processes when choosing the best alternative for development. Moreover, stakeholders can learn and improve their knowledge, as well as gain experience in influencing decision-making processes, when selecting the most appropriate option for development [13,14]. The categories of performance are explained in terms of the following three aspects: procedural, substantive, and transactive.

The procedural dimension means that the assessment complies with acceptable standards and principles and relates to the principles governing impact assessment processes [15]. The consideration of how policies or procedures are being implemented is required and should help frame the methodological dimension, as well as help develop processes based on the implemented techniques [14]. The procedural dimension refers to an evaluation’s conformation with accepted norms and guidelines, as well as its relationship with the guiding principles of impact assessment procedures [15]. It is necessary to take into account the methods used in the implementation of policies or procedures, as this will help define the methodological aspect and build the process using those methods [14].The substantive dimension refers to the achievement of an impact assessment tool’s agreed-upon objectives. The impact assessment tool results in alterations or adjustments to proposed projects or plans [15].The transactive dimension is the ability to employ resources at the lowest cost in the shortest amount of time to achieve the objectives, taking into account the skills and roles of the practitioners. Time and money were also significant components of transactive effectiveness; thus, transactive effectiveness includes resource management in conjunction with impact evaluation, rather than being solely based on the lowest cost [16].

### 2.3. Methods and Tools for the Evaluation of EIAs or EHIAs

Previously, popular checklists have been used for the evaluation of an environmental impact statement (EIS) or a report. An EIS is the output of an EIA that can partially present procedural effectiveness; a good EIS can predict, observe, and design mitigation measures with which to cover environmental problems [17,18,19]. Therefore, to assess the quality of an EIA, most researchers design and develop an evaluation checklist based on a report or EIS. Several modifications of checklists have been proposed and used in many studies, such as the Environmental Statement Review Package, which provides a basic, non-specialist framework for conducting EIS reviews [20,21], the Netherlands Environmental Impact Assessment Commission Operational Criteria [17], the Oxford Brookes University Review Package [19], review criteria [22], and the European Community (EC) guidelines (2001) and packages [23]; however, there is more research that attempts to present an assessment of the quality of EIAs/EHIAs from other perspectives, such as those concerning transactive, substantive, and normative performance [18,19]. The performance of EHIAs has been less frequently evaluated, and there has only been one study specific study performed, on electric power plant projects with regard to procedural effectiveness, developed by Chanchitpricha and Bond (2013) [13]. Nevertheless, prior studies did not employ the identification of changes in the environment and health as evaluation criteria for performance throughout the monitoring period.

### 2.4. Environmental and Health Impact Assessment in Thailand

The environmental impact assessment (EIA) process has been applied in Thailand as a tool for environmental planning and management in the screening approach for economic development projects since 1981. The Office of Natural Resources and Environmental Policy and Planning (ONEP), as well as the Ministry of Natural Resources and Environment (MONRE), are responsible for the administration of EIA and EHIA processes for Thailand. According to the National Environmental Quality Promotion Act (NEQA), 1992, and the Constitution of the Kingdom of Thailand, 2007 (in Section 67, Paragraph 2), Thailand requires projects that may generate an impact on the environment and projects that affect communities severely with respect to the quality of the natural resources, health, and the environment to prepare environmental impact statements (EISs) to be considered for approval before starting said projects [24,25]. The EIA system had been continually developed before the enactment of the NEQA in 1992. This Act had a key role in creating and implementing the environmental impact assessment report review system, which involves a committee of specialists considering the report. The concept of environmental impact assessment system reformation became more concrete when the Kingdom of Thailand’s 2007 Constitution (B.E. 2550) went into effect. The provision in Section 67, Paragraph 2, of the Kingdom of Thailand’s 2007 Constitution (B.E. 2550) inspired the enhancement of the EIA system, such as the requirement for EHIAs. EHIAs are EIAs that incorporate a health component into the evaluation process. According to EHIAs, any project or activity that is thought to have a significant negative impact on a community’s natural resources, health, or environmental quality must conduct a health impact assessment. This assessment must be reviewed and approved by an independent organization that focuses on the environment and health before a permit is issued [26,27,28]. EHIAs comprise a strict public participation standard and the promotion of strategic environmental assessments (SEAs) as a tool for policy–plan–program development, both sectoral and area-based, for executive decision-making.

Since Thailand has used EIAs and EHIAs as tools with which to protect environmental and human health for more than twenty years, the total number of EIAs submitted from the past (1992) to the present (March 2024) is around 11,637 projects, while the total number of EHIAs submitted from 2007 to 2024 is around 121 projects [29]. Most project developments are in the building, residential, and municipal services sector, the energy and renewable energy sector, the industry sector, the transportation sector, and the water resources and agriculture sector.

## 3. Research Approach and Methodology

This study constitutes quantitative research that aims to evaluate the performance of EIAs/EHIAs implemented in Eastern Thailand. There are three steps, in which the research first explores tools for evaluating the performance of EIAs or EHIAs in Thailand and builds drafts of evaluation checklists, then in the second step develops a plausible final evaluation checklist by testing the content’s validity and reliability, which is then finally implemented in a third step. The performance evaluation focuses on three dimensions: the procedural dimension, substantive dimension, and transactive dimension, which cover all stages of EIA/EHIA practices in Thailand. The 13 criteria of evaluation checklists were developed and evaluated by residents around the projects studied, as presented in Table 1.

### 3.1. Study Areas and Selection of Projects

Provinces like Chonburi, Rayong, and Chachoengsao were target markets for the nation’s early development under Thailand’s Eastern Seaboard Development Program because of their geographic strength. Nonetheless, as Figure 1 shows, the province of Chonburi carried out the greatest number of EIA projects between 2007 and 2016 [16,17]. These projects mostly focused on the industrial sector, encompassing transportation, energy, and renewable energy sector projects. Several of these programs were still in the monitoring stage during the study period and were consequently excluded; thus, case study projects were chosen from the Chonburi province for a better and diversified representation of various sectors in terms of EIA/EHIA implementation. All EHIA projects (comprising 100% of EHIA initiatives) that had undergone operation and monitoring were selected, followed by the selection of EIA projects with which to explore performance across different sectors.

Moreso than the performance of EIAs and EHIAs, we were interested in evaluating the difference between EIAs and EHIAs because EHIAs need more information, especially health information, which required more public meetings; therefore, an EHIA project was first selected, and in order to compare EHIAs and EIAs, an EIA project in the same sector was chosen. Additionally, data from other sectors were collected, for considering performance across different sectors.

From 2007 to 2023, there were two EHIA projects that had already been operated and monitored in Chonburi; they were case studies that covered the transportation and industry sectors. Then, in order to explore other sectors, EIAs were added to the study, which were EIAs in the energy and renewable energy sector. Four EIAs were selected because we were interested in studying the differences in the performance of EIAs and EHIAs; therefore, the number of EIAs was not too different. Therefore, all EHIAs (100% of EHIAs) that had already been operated, monitored, and selected, as well as the EIAs, were selected for examining the variance in performance over different sectors.

Ultimately, this analysis selected four EIA projects: one for the industry sector (EIACB3), one for the transportation sector (EIACB1), one for the estate sector (EIACB4), and one for the energy and renewable energy sector (EIACB2). One EHIA project was chosen for the industry sector (EHIACB2) and one project for the transportation sector (EHIACB1).

### 3.2. Data Collection

In order to assess performance using a criteria checklist that was designed to focus on three dimensions—procedural performance, substantive performance, and transactive performance—primary data were gathered through interviews and field surveys with villagers, local fishing groups, non-government organizations, the local authorities responsible for EIAs and EHIAs, and municipality as well as medical personnel in sub-district health-promoting units. In order to consider the EIS database and monitoring reports, secondary data were gathered via the ONEP website [32].

#### 3.2.1. Primary Data

The sample for this study was identified from the comprehensive list of communal households provided in the EIA and EHIA reports. It encompassed a total of at least 162 individuals residing in households across the province of Chonburi. To select participants, the researchers employed a random sampling method, targeting households where at least one member had attended public meetings pertaining to EIAs and EHIAs, because they have information about or experience of the stages of producing and gathering data for EIA and EHIA reports. If no individuals within a household had participated in public meetings concerning EIA and EHIA processes, we then continued to select new individuals from other families randomly in the same places, until individuals with relevant meeting experiences were identified.

Considering relevant meeting experiences, we identified a total of 162 individuals as key informants living in the vicinity of EIA and EHIA projects, who assessed the performance based on their interactions with the case, monitoring report, and EIS. In order to generate quantitative data on project performance using a rating scale, a questionnaire was employed in surveys completed by the sample. The evaluation assessed the EIAs’ and EHIAs’ quality, efficacy, and follow-ups using the checklist that we created from the relevant research regarding the criteria.

#### 3.2.2. Secondary Data

The information on EIA and EHIA projects details baseline information about the environmental, social, and economic conditions in the project area, the impact assessment results, measures for mitigating impacts as well as impact management, and environmental management plans. The monitoring reports were sent to the related responsible organization agencies regularly, two times a year, to monitor and evaluate trends in situations that affected the environment after the project was developed or implemented, and were also used to improve or increase efficiency in complying with environmental measures. Both EIA reports and monitoring reports were collected as electronic files through the ONEP website [25].

### 3.3. Data Analysis Methods

The data used for the analysis of the performance only calculated the answers of respondents who had experienced the EIA/EHIA process and had joined a public meeting. The performance types to be assessed were procedural, substantive, and transactive. The performance was assessed via a rating score (needs improvement = 1, fair = 2, and excellent = 3) for assessing the substantive and procedural performance, while another dimension (transactive performance) used insufficient = 1 and sufficient = 2. Finally, a multivariate analysis of the dataset (categorizing the studies by attributes) was carried out and represented the three dimensions of performance by identifying the percentage of performance scores, interpreting them to be excellent, good, just satisfactory, poor/just unsatisfactory, and very poor/unsatisfactory. The overview of performance was presented as follows: 100% of the total score meant “Excellent”, 75% to under 100% of the total score meant “Good”, 50% to under 75% of the total score meant “Just satisfactory”, 25% to under 50% of the total score meant “Poor/just unsatisfactory", and 0% to under 25% of the total score meant “Very poor/unsatisfactory”. 

For example, for the transactive performance, the evaluator needed to consider three transactive criteria. The possible max total score of transactive performance is six, equal to 100%. Therefore, if the evaluation result is five, the performance is equal to 83.33%, and is interpreted as a “Good” performance. The differences in EIA and EHIA performance were analyzed by employing Statistical Package for the Social Sciences (SPSS ver. 20) software.

## 4. Results

### 4.1. Characteristics of Selected Projects

The cases studied consisted of a variety of sectors in both EHIA and EIA projects; there were two EHIAs in the transportation sector (EHIACB1) and one in the industry sector (EHIACB2), as well as four EIAs, of which one was in the energy and renewable energy sector (EIACB2), one was in the industry sector (EIACB3), which specifically entailed aluminum and steel production, one was in the transportation sector (EIACB1), and one was in the estate sector (EIACB4). Both transportation projects included the port, with similar activity but different sizes; therefore, the big project was an EHIA and the other one was an EIA.

### 4.2. Respondents’ Characteristics

The 162 respondents were divided into nine local authorities (five in the administrative department and four in the health department) and 153 villagers. The respondents were people who had information and experience about a project, from the stage of assessing the impact, identifying the impact, and also considering mitigation measures as well as the monitoring phase. They could understand and see the growth of a project step-by-step. They were the key stakeholders of the EIA and EHIA processes. They could be representative of the total population or stakeholders of the EIA and EHIA processes, because the respondents were selected with no bias and had an equal chance of being chosen. They were not related to the six projects analyzed below. Most of the respondents were in the age category of 31–40 years (71.0% of all respondents). Almost all of the respondents were government officers (91.4% of all respondents), covering persons whose work was related to EIA and EHIA processes in the areas of the cases studied; there were four from the local health organization, one from the local environment and resources, and four from the municipality, as shown in Table 2. More than 80% of respondents had lived in the area for approximately 5–10 years and were therefore able to present and be aware of the environment and recent health changes as well as impacts. The survey results also demonstrated that 45% of the respondents had experience in the EIA and EHIA monitoring phase, 11% had participated in EIA and EHIA monitoring committees, 45% had the opportunity to visit a factory in the implementation process to learn about the technologies and measures used for treatment and control pollutants, and 38% observed and surveyed, through the environmental situation board, the air quality, Volatile Organic Compounds (VOCs) in the air, and the water quality displayed around their community.

### 4.3. The Performance of EIA and EHIA Practices

The results indicate that all of the cases studied (six cases) were just satisfactory (100%), with no case rated as either excellent or very poor. The results show that the performance of EIA/EHIA practices in Chonburi province revealed a relatively good picture of the overall performance. The result indicates that 83% of all cases studied (five of six cases) were of just satisfactory performance; the average total score was 64%. The lowest score was 57% of the total score and the highest score was 81% of the total score. The cases were “Just satisfactory” in terms of substantive performance (72% of the total score average), procedural performance (59% of the total score average), and transactive performance (77% of the total score average). There was no statistically significant difference between cases. The EIACB2 case had the best performance overall, at 81% of the total score, and also had excellent scores in all of the transactive criteria (Table 3). The performances of the studied EHIA cases and the studied EIA cases were not statistically different in all dimensions of performance, as presented in Table 4.

#### 4.3.1. Substantive Performance

Substantive performance was assessed through the following three criteria checklists (S1–S3) concerning the main objectives of EIAs and EHIAs: whether the project proponent or the consulting firm was implemented, using the EIAs and EHIAs to assess the impact of environment cover on both positive and negative results, and mitigation measures were identified (S1); whether the opinions of the public were taken into account in the decision-making process (S2); and whether the EIA and EHIA reports presented enough information for decisions (S3). The substantive performance of all project cases studied presented a high in-range performance score. As presented in Table 5, five of the case studies were just satisfactory performances, while one project case was a good performance. The average scores were 69% of the total score in S1, 73% of the total score in S2, and 66% of the total score in S3.

#### 4.3.2. Transactive Performance

The transactive performance was assessed in the supporting section of the EIA/EHIA process, with a focus on three primary resources: human resources, financial resources, and time. The results of this assessment showed a range of scores, from satisfactory to excellent, in terms of transactive performance (as shown in Table 6). More details on the scores can be found below.

The time used to carrying out EIA and EHIA reports within a reasonable time frame without an undue delay (T1) for all of the projects received a high range of scores. Three of the projects were just satisfactory, two of them had good performance, and one of them showed excellent performance. The results showed that all of the projects were able to manage their time effectively; however, some projects required additional details in the report. It is worth noting that the report was prepared and approved before the construction and operation of the projects.

The results for the specification of roles (T2) indicated that all of the projects had the highest average score for transactive performance, which totaled 78%. The EIACB2 project achieved the best score and was rated excellent in T1–T3. When considering performance across different criteria, EIACB1, EIACB3, and EIACB4 (the cases in the transportation, industry, and estate sectors, respectively) were rated as just satisfactory. On the contrary, EIACB2 (the case in the renewable energy and energy sector), EHIACB2, and EHIACB1 (the cases in the transportation and industry sectors) were rated excellent and good, respectively.

Regarding the last criterion of financial resources (T3), the main sources of funds for impact assessment processes and permission approvals came from project developers or proponents. Additionally, the Department of Environmental Quality Promotion in Thailand provides financial support for public consultations. As a result, the evaluation score for all cases was just satisfactory, with an average score of 74% out of the total possible scores.

#### 4.3.3. Procedural Performance

In the study of six cases, three procedural performance criteria (P5, P6, and P7) were found to be satisfactory and good overall; however, EIACB1’s procedural performance criteria (P1–P4) were poor. The study found that procedural performance was just satisfactory in 83% of cases. The average percentage of the total score for all cases studied was 59%. The procedural performance score ranged from 46% to 69% of the total score. The P6 criterion had a significant proportion of procedural performance as satisfactory performance (as shown in Table 7).

Of the total cases studied, 83% and 67% were just satisfactory in terms of policy frameworks and procedures for EHIA and EIA processes (P1) as well as in terms of institutional characteristics (P2). For delivering reports to participating stakeholders for review (P3), EIA/EHIA regulation considers public consultation a priority in the EIA/EHIA process. It is mandatory, and a public participation process was conducted in all of the cases studied. Thus, almost all cases meet just satisfactory performance in this criterion (67% of the cases studied); however, the score of P3 was the lowest in the criteria of procedural performance.

In terms of the reliability of data in environmental impact assessment reports (P4), 67% of cases studied had just satisfactory performance, which was the same as the reliability of data in the environmental impact assessment monitoring report (P5) criterion; all of the cases studied were the best-performing criterion, as the highest average percentage of the total score was 65%.

Considering the criterion performance of the control and implementation of measures specified in work plans (P6), all of the cases studied were just satisfactory. This criterion was in line with the P5 results; the public had the opportunity to participate in committees to assess and consider how factories or project proponents implemented mitigation measures, as well as to consider information about the waste generated and pollutant emissions under the national pollution control standard [24,33] through the monitoring report.

For the recognition of environmental and health impact changes (P7), the results showed that almost all cases were just satisfactory (83% of all cases studied). The recognition of environmental and health information showed that the worse changes clearly affected the physical environment, i.e., the damage of and the accidents on the road, the utilization of public water resources that were directly related to the factories expanding in the area, and deficiency predictions about the cumulative impact and utilization of changes and problems.

## 5. Discussion

The findings indicate that there were no statistically significant differences in the performances of different initiatives. This was in line with Chanchitpricha and Bond [34], a study on the procedural performance of power plant projects in Thailand that found similar results regarding satisfaction. The renewable energy and power project had the largest percentage score for procedural performance and had a simple satisfactory performance. When one looks at procedural performance in other industries, like transportation, manufacturing, and real estate abroad, the results align with the majority of earlier research, which demonstrated satisfactory performance and above-average scores [35,36]. Impact assessment organizations have uneven management capabilities in EHIA and EIA practices because of the various features of various departments and regions, particularly in complex scenarios [37].

For P1, Thailand has the guidelines and procedures covering all the project’s sectors that are implemented in line with provincial economic development policy, but the unclear and inadequate nature of the guidelines is an essential factor influencing environmental and health impact assessments. Although EHIAs have clearly defined procedures and guidelines, in practice, due to the different characteristics of different regions and departments, and especially complex situations, impact assessment organizations have inconsistent management and, in some cases, the quality of impact reports cannot be effective [37,38].

Institutional roles, collaborations, and infrastructure (P2) can become more transparent for all pertinent stakeholders and authorities when legislation is implemented. Thailand networking in EIA/EHIA systems, such as data sharing between organizations, is quite low [37], the library of electronic files of EIA/EHIA reports is not 100% accessible and updated, and the level of information sharing between local authorities is low.

According to EIA/EHIA procedures, the project proponents or consultant firms have to publish or deliver reports to participating stakeholders for review (P4), in order to collect information about the appropriate way to communicate and receive information about projects; however, in practice, the main way that project proponents or consulting agencies make their reports publicly available is by providing only one hard copy of the report or summary report to the head of a village, while public observations within the public participation process should be submitted online only. Moreover, people can access the hard copy of the final report at the local authority, the permitting agency, and at the ONEP center, while it is also published online on the ONEP website; however, this is not suitable for all people, and it is especially detrimental to populations of rural areas where there is no infrastructure and technological limitations (i.e., access to the Internet). Thus, villagers may not know some information about a project and not have the opportunity to comment or verify the completeness and accuracy of the information in a report [8,37]; participation can hardly be effective if people cannot even access information on projects that they are expected to give opinions on.

The reliability of the information in EIA/EHIA reports is one of the criteria with a low performance score; most respondents cannot access the final version of a report before approval. Previous research [39,40] has shown that the public lack information and transparency and are uncomfortable about expressing their concerns, and that reports were made without the feedback loop being completed.

Every case that was examined had clear definitions, assigned responsibilities that were completed by the most qualified individuals, and the development of the knowledge and labor needed for EIAs and EHIAs in accordance with an EIS, the person in charge, and their position. This surpasses regulatory requirements due to the significant significance and uniqueness of the relevant authorities and individuals [41]. Thailand has limitations in terms of professional and technical personnel, impact assessments of human resources, skills, and capacity, particularly local human resources, much like other developing nations [4,39,42]; however, the amount of employees needed to cover all stages of EIA and EHIA processes, as well as the numerous subcommittees required during the monitoring phase, provide challenges for the local authorities in Thailand.

Regarding public participation, the average percentage score indicated that all initiatives were completed with just adequate performance; however, when looking at the cases examined, it is discovered that the EIACB1 transportation sector project had the lowest percentage of impact assessment reports’ total score for public reliability, as well as a low score for delivering reports for public review. These findings are consistent with the findings of Chompunth C. [43], which suggest that no law gives the public the authority or role to make decisions and that the public does not have enough influence over decision-makers’ decisions or access to a sufficient degree of participation in decision-making. Furthermore, the public consultation was conducted using ambiguous guidelines, such as the requirement that information be posted publicly on announcement boards at the local and federal levels of administration (without specifying which channels should be suitable and appropriate for the community) and the requirement to disclose information for sufficient periods of time (without specifying the minimum time period) [44]. Similar to the findings of Suwanteep et al. [45], there are still contentious concerns regarding short public involvement periods and restricted access to EIA and EHIA project material in actual public engagement practices. Additionally, Thailand is working to improve public accessibility by publishing reports online to support the public in fulfilling their right to information access, even though stakeholders only have limited access to the Internet due to a lack of essential information [41,46]. Thus, in various situations, the audience should be compatible with the proper communication and methods of information delivery.

The information revealed during the project’s implementation stage, as shown in criterion P5, which is consistent with the documentary reviews and highlights the laws for post-EIA activities in EIA monitoring and auditing activity, also increased the reliability of EIA and EHIA practices in Thailand. The consulting firm submitted monitoring reports every six months and the facility was visited for surveillance and monitoring meetings within the factory to demonstrate their sincerity and provide the public and pertinent organizations with proof of the data’s dependability. Furthermore, Thailand boasts an online system that the general public can access and check at any time to monitor environmental quality, receive results in real time from government and private-sector partners, and maintain an environmental monitoring network [34]. For example, EIACB4 is the case in the estate sector that established the Environmental Monitoring and Control Center (EMCC) to continuously monitor environmental quality and air quality both in the ambient air and air emitted in the area.

## 6. Conclusions

The overall performance of the instances examined shows that none of the cases fell into the extremely good or extremely poor categories; instead, all of the cases performed at a level that was just satisfactory. There was no statistically significant difference in EHIA and EIA performance. Transactive performance accounted for the largest percentage of the overall score (65.4% ± 18.93 in EHIAs and 69.4% ± 21.51 in EIAs), followed by procedural performance (49.8 ± 16.36 in EHIAs and 52.6 ± 17.13 in EIAs) and substantive performance (47.4 ± 17.63 in EHIAs and 51.2 ± 18.51 in EIAs), in that order. Public involvement received a poor score in two criteria, indicating that it is one of the critical issues that need more sufficient attention. The majority of EHIA and EIA cases can be accessed online through monitoring reports, which are constantly updated; however, EISs are more challenging to obtain in both hard copy and electronic formats.

This study’s primary innovation is its adaptation of a commonly-used assessment methodology from earlier research, casting stakeholders with firsthand, project-related experience in the role of the evaluator; however, because this study had to evaluate and encompass all phases of EIAs and EHIAs, the respondents selected therefore had the limitation of representing a variety of groups. The study’s findings indicate that the projects’ percentage of public participation and reliability scores were low. Consequently, to improve performance, it is recommended that an open access database be created and updated on a regular basis. This database would allow interested stakeholders to participate, voice their opinions, and double-check data at any point during the project, as well as to follow up on and monitor projects that were implemented to foster public trust, transparency, and dependability. Moreover, the agencies in charge of EHIAs and EHIAs carry out a cycle of systemic improvement, incorporating knowledge co-production, regular report quality reviews, and periodic monitoring.

## Figures and Tables

**Figure 1 ijerph-21-00644-f001:**
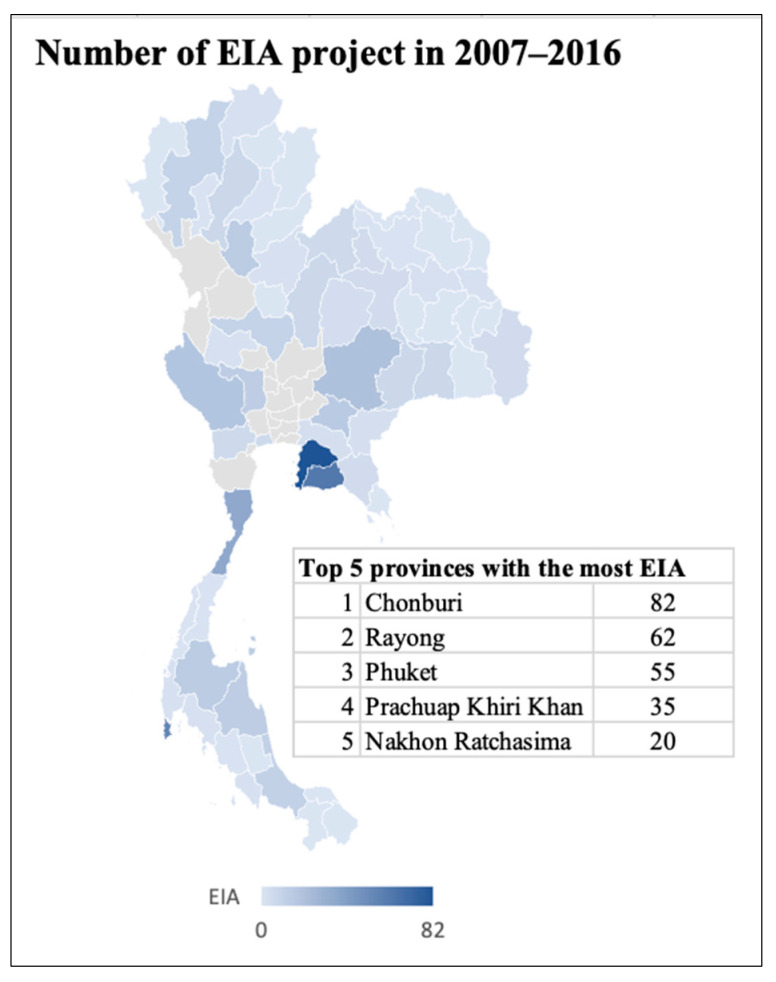
The number of EIAs implemented between 2007 and 2016 used by the authors; data were taken from [29,30,31,32]. Deep blue shows the location of Chonburi, followed by Rayong and Phuket.

**Table 1 ijerph-21-00644-t001:** The evaluation checklists developed.

**S**	The substantive performance.
**S1**	The project was implemented under the objectives of an EHIA/EIA.
**S2**	The management has given importance to and brought opinions from the public when considering the project in the decision-making process.
**S3**	Presentation/information specified in the report is sufficient for the decision to develop or suspend the project.
**P**	The procedural performance.
**P1**	Relevant policy framework and procedures for the EHIA/EIA process.
**P2**	Institutional characteristics—institutional roles, infrastructure, and collaborations of relevant authorities.
**P3**	Delivering the report to participating stakeholders for review.
**P4**	Reliability of the data in an environmental impact assessment report.
**P5**	Reliability of the data in an environmental impact assessment monitoring report.
**P6**	Performance of the control and implementation of measures specified in the work plan.
**P7**	The recognition of environmental and health impact changes.
**T**	The transactive performance.
**T1**	Time—EIA and EHIA reports were carried out within a reasonable time frame without undue delay.
**T2**	Specification of roles and responsibilities were clearly defined and allocated, tasks were undertaken by the most appropriate subjects, and the acquiring of skills and personnel required for the EIA and EHIA was completed.
**T3**	Financial resources—carrying out the EIA and EHIA did not entail excessive spending.

**Table 2 ijerph-21-00644-t002:** Results of the public responses.

Characteristics of Respondents	Frequency	Percent
Age		
20–30	32	19.8
31–40	115	71.0
41–50	11	6.8
51–60	4	2.5
Occupation		
Employee (inside the factories or estate)	2	1.2
Merchant	4	2.5
Self-employed	8	4.9
Government officer	148	91.4
Time spent living in village		
Less than 5 years	39	24.1
5–10 years	111	68.5
More than 10 years	12	7.4
Experience in EIA/EHIA monitoring/monitoring and surveillance of environment and health		
Site visited	33	45.2
On committees	8	11.0
Observed from the environmental and health status board displayed in real time	31	38.3
From monitoring reports	8	9.9
Website of organizations of pollution control and related	1	1.2

Source: questionnaire survey of local communities (2023).

**Table 3 ijerph-21-00644-t003:** The overall performance of EIA and EHIA practices divided by criteria.

	% of the Total Score					
	EHIACB1	EHIACB2	EIACB1	EIACB2	EIACB3	EIACB4
S1	63	78	62	73	63	74
S2	63	85	66	77	71	78
S3	56	62	63	70	77	68
T1	56	84	46	100	43	45
T2	81	83	63	100	71	75
T3	80	93	43	100	66	60
P1	52	58	46	68	69	55
P2	49	52	44	67	67	57
P3	44	55	43	67	69	56
P4	44	52	38	68	69	51
P5	62	63	56	68	76	63
P6	58	65	50	72	69	56
P7	52	64	46	75	64	60
Overall score	64	75	57	81	70	62

Source: questionnaire survey of local communities (2023).

**Table 4 ijerph-21-00644-t004:** Comparison of means of EHIA and EIA performance by dimensions of performance.

Performance Type	Mean of Percentage of Total Score (with Standard Deviation)		*t*-Test	*p*-Value
	EHIA	EIA		
Substantive	47.7 (17.63)	51.2 (18.51)	−1.861	0.063
Transactive	65.4 (18.93)	69.4 (21.51)	−1.934	0.054
Procedural	49.8 (16.36)	52.6 (17.13)	−1.615	0.107
Overall performance	55.4 (18.79)	58.1 (18.58)	−1.442	0.150

Source: questionnaire survey of local communities (2023).

**Table 5 ijerph-21-00644-t005:** Substantive performance results.

Performance Level	Substantive Performance	Percent of Project		
		S1	S2	S3
Excellent	0	0	0	0
Good	One project	17	50	17
Just satisfactory	Five projects	83	50	83
Poor/just unsatisfactory	0	0	0	0
Very poor/unsatisfactory	0	0	0	0

Source: questionnaire survey of local communities (2023).

**Table 6 ijerph-21-00644-t006:** Transactive performance results.

Performance Level	Transactive Performance	Percent of Project		
		T1	T2	T3
Excellent	One project	20	20	17
Good	Three projects	20	50	33
Just satisfactory	Two projects	60	40	33
Poor/just unsatisfactory	0	0	0	17
Very poor/unsatisfactory	0	0	0	0

Source: questionnaire survey of local communities (2023).

**Table 7 ijerph-21-00644-t007:** Procedural performance results.

Performance Level	Procedural Performance	Percentage of Project						
		P1	P2	P3	P4	P5	P6	P7
Excellent	0	0	0	0	0	0	0	0
Good	0	0	0	0	0	17	0	0
Just satisfactory	Five projects	83	67	67	67	83	100	83
Poor/just unsatisfactory	One project	17	33	33	33	0	0	17
Very poor/unsatisfactory	0	0	0	0	0	0	0	0

Source: questionnaire survey of local communities (2023).

## Data Availability

Data are contained within the article.

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
