# Peer review of "The Performance of Environmental and Health Impact Assessment Implementation: A Case Study in Eastern Thailand"

_ijerph, 2024, doi:10.3390/ijerph21050644_

Round 1
Reviewer 1 Report
Comments and Suggestions for Authors
Comments on the Quality of English LanguageAuthor Response
Please see the attachment.

Reviewer 2 Report
Comments and Suggestions for Authors
The authors develop in this paper an analysis of the performance of EIA and EHIA in Thailand through the case study of six projects in the Chonburi area. The paper is well written, presents the justification of the theoretical framework, develops the application methodology, presents the results and discussion, and the conclusions are in line with the findings of the research.
The article is interesting, and it is of interest in this type of study where the effectiveness of tools such as EIA and EHIA is shown through the interaction with users in the area and who were involved in the process, although I consider that it should have the following indications to be suitable for publication.
1.- The EIA tool is widely known, but the EHIA is not so well known, so it would be useful to have a better explanation of what it consists of.
2.- The study area (Chonburi) was selected because it is the area with the highest number of projects in the period 2007-2016, a total of 82 projects, and then only 6 projects were analyzed. This should be better explained. I understand that the survey is done in 2023, but why no projects have been included between 2016 and 2023, or at least until 2022. The analysis data may not reflect the current reality of the procedures. This point should be justified because it seems to be one of the most important weaknesses of the design of the procedure.
3.- Why does the survey comprise 162 individuals? How representative are they of the total population or stakeholders of the EIA and EHIA processes? and are they related to the six projects analyzed later?
4.- Lines 185 and 186 state "The data used for the analysis of the performance calculated only the respondent's answers who had experienced in EIA/EHIA process and never joined in the public meeting". From Table 2 it is understood that there are 81, is this correct? They should be indicated in the text.
5.- Results. The results of the surveys of the first part of the methodology are presented but the results of the analysis of the EIA-EHIA reports are not indicated as clearly, I think that at least one example for each of the dimensions (procedural, substantive, and transactive) would help to understand and justify more the evaluation made.
6.- Discussion: What is the possible explanation for the poor procedural performance of the criteria P1-P4?
7.- In the Discussion or Conclusions it would be interesting to point out the limitations of the article, as well as future lines of research.
Minor aspects
1.- Abstract. Line 16. EHIA should be described before the acronym is used.
2.- Lines 187-190. The wording should be revised. There are punctuation errors.
3.- Lines 207-208. Idem. The wording should be revised. There are punctuation errors.
Round 2
Reviewer 1 Report
Comments and Suggestions for Authors
I am satisfied with the current version of the manuscript.
Reviewer 2 Report
Comments and Suggestions for Authors
My indications have been taken into account and the understanding of the research has been improved.
Many thanks to the authors for their efforts.